# On the Determinants of Green Technology Diffusion: An Empirical Analysis of Economic, Social, Political, and Environmental Factors

Busra Agan [1] and Mehmet Balcilar [1,2,*]

1   Department of Economics, Eastern Mediterranean University, Via Mersin 10,
    Famagusta 99628, North Cyprus, Turkey; busra.agan@emu.edu.tr
2   Department of Economics, OSTIM Technical University, Ankara 06374, Turkey
*   Correspondence: mehmet@mbalcilar.net

**Abstract:** The purpose of this study is to investigate the importance of economic, social, political, and environmental factors in determining green technology diffusion. We use a unique annual panel dataset covering 58 nations from 1990 to 2019. Based on dynamic panel data models estimated using system generalized method of moments (GMM), we test whether the technological achievement of nations, general technology diffusion trends, environmental performance, democratic accountability, income distribution, foreign direct investment, income level, and socioeconomic conditions are significant determinants. Our findings indicate that green technology diffusion has a significant relationship with all of these factors. We obtain new evidence that general or brown technological innovation, diffusion, and achievement trends in a country are significant drivers of environmental technology diffusion. The findings of our paper have significant implications for sustainable development, given the importance of green products and technologies. The results of the study suggest that policies aimed at promoting the diffusion of green technologies may not be successful in the presence of unfavorable economic, social, political, or environmental conditions.

**Keywords:** green technology diffusion; climate change; sustainable development; technological achievement; dynamic panel data

## 1. Introduction

The benefits of the creation of technology and technology diffusion in climate change mitigation have been greatly emphasized in recent policy debates in academia and policy making institutions (e.g., [1–4]). The achievement of environmental goals for sustainable development, as well as the minimization of their costs, are contingent on the development and diffusion of innovative, environmentally friendly technology. Market failures in terms of technological development are conceptually distinct from environmental externalities, implying that the deployment of new, ecologically beneficial technology suffers from a "double externality," making it a significant topic of interest for researchers and policy makers.

Although the creation of green technologies plays a key role in climate change mitigation and sustainable development, these technologies may not be adopted widely and their benefits may not be realized. New technologies become beneficial only when they are widely adopted, which is governed by the process of technology diffusion. Schumpeter [5] defines the technological progress as consisting of three stages: invention, innovation, and diffusion. The benefits of a new technology become widespread through the process of diffusion. As Allan et al. [4] points out, the diffusion of green technologies is even more important than their invention and innovation for climate change adaptation. The distinguishing feature of a "green" technology is that it generates or facilitates a reduction in environmental externalities relative to the status quo. In comparison to the status quo,

"green" technologies generate or promote a reduction in environmental externalities. According to some analyses, the widespread use of such technology will result in enormous benefits [2,4]. Economists are skeptical of such assertions, pointing out that evaluations usually overlook factors of that may impede the diffusion of new technologies.

Waisman et al. [6] provided a perspective that dives into the implementation contexts, systems, and enabling conditions of green technology implementation in order to identify actions that can more effectively foster, accelerate, and enable technology adoption. As a result, this approach assists governments in identifying steps that will expedite, encourage, and allow technological adoption, as well as in strengthening the policies, strategies, and legal frameworks that support those efforts. The perspective brought forward by Waisman et al. [6] is broadly linked to the point raised by previous literature that the benefits of green technology are contingent on its widespread diffusion. Given the issues of reducing global warming and mitigating the effects of climate change, even national-level actions will have a limited impact, implying the need for global policy frameworks. There may be micro-level reasons for the slow diffusion of green technologies, such as consumer preferences and/or costs, in addition to the obvious installation costs adopted by the public [7–9]. However, macro-level considerations provide the framework for determining whether green technology adoption will be broad enough to generate environmental advantages.

Against this backdrop, this study examines a broad range of macroeconomic factors at the country level that may have an effect on the diffusion of green technology. Although some of these characteristics, such as the overall trend in technology diffusion and technological achievement, have not been examined previously, there are theoretical reasons to believe they are significant to green technology diffusion. We evaluate the following aspects as potential causal factors for green technology diffusion: technological achievement, general trends in technology diffusion, environmental performance, carbon dioxide ($CO_2$) emissions, democratic accountability, income distribution, income level, education as a measure of human capital, foreign direct investment, and socioeconomic conditions. Several of these factors, such as $CO_2$ emissions, inequality, and democracy, have been researched in the contexts of environmental sustainability but not specifically as determinants of green technology diffusion (see, for example, [8,10–16]). To the authors' knowledge, no other study has evaluated the influence of these factors on the diffusion of green technologies.

As pointed out by Allan et al. [4], many of the characteristics of the technology diffusion process for green technologies are similar to those of general technology diffusion. There is a substantial body of literature on general technology diffusion. Allan et al. [4] survey green technology diffusion in relation to general technology diffusion and review the related economic theory. Therefore, we consider the general technology diffusion trends in a country as one of the factors determining the adaptation rate of green technologies. It often takes a long time for new, intrinsically superior technologies to diffuse widely, and diffusion rates vary greatly. For example, while it took 40 years for the clothes washer to go from one quarter of households to three-quarters, color television achieved the same amount of diffusion in less than 10 years [2]. However, the technology diffusion in a country is interlinked to the creation of new technologies or innovation, the diffusion of old and new technologies, and human skills [17,18]. These indicators serve as a measure of a country's "technical advancement". As a result, we construct a technical achievement index and investigate its relevance to the diffusion of green technology.

Popper [19,20] asserts that democratic social arrangements are more conducive to innovation. The scant empirical evidence in this subject casts doubt on democracy's positive role on innovation [15,21,22]. To our knowledge, however, no research has examined the impact of democracy on environmental technology diffusion, which we anticipate will have a different dynamic than technology innovation and other type technology diffusion. Income distribution is another factor that may affect the diffusion of environmental technology. As noted by Magnani [23] and Pfaff [24], inequality has a complicated impact on the creation and diffusion of green technologies. When the median voter's income surpasses

the national average, economic redistribution increases the demand for environmental protection. The per capita income is another factor, as most of the research and development (R&D) leading to innovations and its adaptation happen in high-income countries. Education is critical to a country's progress. Bartel and Lichtenberg [25] believe that highly educated individuals are more inclined to adopt new technologies, hence influencing their diffusion and demand. Foreign direct investment (FDI) flows, as argued by Driffield and Love [26], Popp [27], and Dechezleprêtre et al. [28,29], may function as critical drivers of environmental technology creation and diffusion, compelling us to investigate them as a factor impacting green technology diffusion. In worsening environmental conditions, such as higher $CO_2$ emission, better socioeconomic conditions and higher environmental performance enhance the demand for environmental quality, hence, they also impact environmental technology innovation and diffusion in a positive way [30–32].

The objectives of this paper are threefold: (i) to examine the economic, social, political, and environmental factors of green technology diffusion in 58 countries, (ii) to fill an important knowledge gap by examining whether the general technology diffusion trends and technological achievements of nations are also the source of trends in green technology diffusion, and (iii) to explore several dynamic panel data models using the system GMM method and show that these factors do indeed play a significant role in the diffusion process of green technologies. The results are robust to model specifications.

By incorporating additional dimensions of green technology diffusion, this study contributes to the existing literature in several ways. First, unlike prior research, we place a premium on the distribution of green technology, rather than on its creation. Second, we enhance past research by offering a comprehensive set of criteria that may affect the diffusion of green technologies. While the majority of countries have made only a few advances in green technology, the spread of green technology has reached practically every country. Green technology's spread should not be considered in isolation from a country's technological achievement. Third, along with other factors, green technology is likely to be influenced by brown technical change. Our paper studies green technology dissemination in relation to nations' technical accomplishment, which encompasses both the generation and spread of new technologies. Fourth, we also determine whether brown technology—which covers a much broader range of technologies—diffusion trends are a significant driver of green technology diffusion, thus, filling an important knowledge gap in the literature. Fifth, we extend previous research by incorporating a broad range of factors that may influence the diffusion of green technologies, including democratic accountability, per capita income, education, socioeconomic status, foreign direct investment, $CO_2$ emission, and environmental performance. These variables comprise the most comprehensive set of variables, not all of which have been explored previously. Sixth, we update the technological achievement index of Desai et al. [17] from 1990 to 2019 and expand its scope to 58 nations. Thus, in comparison to earlier research, our study covers a longer time period and includes a greater number of countries. Seventh, we also estimate panel data models that account for the endogeneity of some explanatory variables. Additionally, unlike earlier research, we estimate dynamic panel data models for a longer period, which allows for sluggish adjustment and lagged effects to be taken into account. Our empirical findings demonstrate that these factors are good determinants of the diffusion of green technologies. The evidence is unaffected by a variety of mode specifications. Our findings have important policy implications, since they imply that unfavorable country-level macro conditions may impede policy initiatives targeted at promoting green technology development and adaptation.

The remainder of the paper is structured as follows: Section 2 conducts a review of the relevant studies. Section 3 discusses the empirical methodology and explains the data used in the study. Section 4 analyzes the empirical findings, and Section 5 concludes.

## 2. Literature Review

Global warming is a significant issue in the modern era, with a variety of consequences for humanity and the environment. Climate change is a factor in global warming. Due to

the underlying threat to the planet's energy balance and climate, technological innovation must be encouraged.

A number of studies examine the relationship between democracy, inequality, and country-level environmental technology change. Scruggs [33] investigates the effects of economic and political inequality on environmental regulations; his findings indicate that increasing inequality can result in a degradation of environmental quality, as the sustainability approach assumes. Vona and Patriarca [13] examine the relationship between inequality and green technologies for the members of the Organisation for Economic Co-operation and Development (OECD) between 1985 and 2000, concluding that inequality in rich countries is negatively correlated with the diffusion of green technologies, whereas inequality in poor countries is dependent on per-capita income. Kempf and Rossignol [10] analyze the relationship between inequality and green technology policy using a theoretical approach in which they model the endogenous growth process. Their theoretical conclusions imply that increasing social inequality has a detrimental effect on the ecosystem. Even in the face of adverse environmental conditions, an unequal society tends to consume more resources. Nonetheless, You et al. [34] assess the influence of democracy and financial openness on $CO_2$ emissions using data from 1985 to 2005, finding evidence that greater democracy appears to cut emissions, while increased financial openness appears to have no effect on emissions. Lv [35] studies the influence of democracy and income on $CO_2$ emissions data from 1997 to 2010 in 19 developing nations, finding evidence that democracy reduces emissions when a country's wealth level is above a specific level. Zecca and Nicolli [15] evaluate the factors and advancements of environmental technologies and eco-friendly innovations using data between 1980 and 2013, utilizing a dataset of patent applications, inequality, and democratization on green technological transformation. Economic growth, inequality reduction, and more democratic societies all have a substantial association with environmental technological change in 77 nations, according to their empirical findings.

Additionally, the literature abounds with survey studies on the diffusion and development of green technologies. Allan et al. [4] conduct a review of the literature on the dissemination and development of green technology, focusing on policy instruments as well as the adoption decisions of individuals and businesses. Their findings indicate that certain elements such as knowledge, awareness, cognitive biases, and values all play a significant influence in the diffusion of technology. Popp et al. [11] conduct another review of the literature, focusing on the role of green technical advancements. Their survey shows that the process of developing environmentally friendly policies encountered numerous obstacles, including resource constraints, climate change adaptation, R&D investment, and environmental regulation. This innovation process, on the other hand, establishes a link between the adoption of new technologies, the diffusion of knowledge, and the long-term sustainability of economic growth.

The majority of the existing research on green technological diffusion in the literature has focused on empirically measuring cross-country technological innovation and diffusion. Hall and Helmers [36] examine significant innovators' patents on green technologies, utilizing the OECD countries' green technology patent classes. Their findings indicate that the pledged patents serve to ensure the safety of environmentally friendly innovations and aid in their diffusion. Nonetheless, Jin [37] studies worldwide technology diffusion using data from the OECD's multi-region general equilibrium model. The findings indicate an increase in the dissemination of green innovations, cross-country R&D activities, and incentives for global climate mitigation. Du et al. [14] evaluate the influence of green technological diffusion on $CO_2$ emissions throughout the 1996–2012 period using a panel dataset covering 71 nations. These effects are dependent on countries' income levels; the higher the income level, the more economies contribute to $CO_2$ emission reductions. Recently, Halkos and Skouloudis [38] published another study analyzing the diffusion and development of environmental technology utilizing a country-level database for 56 nations from 2005 to 2014. Their findings indicate that government effectiveness, foreign direct investment, and human capital development all contribute to the spread of environmental technolo-

gies. Lv et al. [39] use the data envelopment analysis (DEA) methodology and the global Malmquist-Luenberger index (GML) to examine the relationships and disparities between financial scale, financial structure, and financial efficiency on green technology innovation in China from 2003 to 2017. The empirical findings reveal that financial efficiency and financial scale have a detrimental effect on green technology innovation, whereas financial structure has a beneficial effect. Additionally, Horbach [40] examines the causes of green technology dissemination using a German firm-level survey dataset, obtaining evidence that the growth of inventions, environmental instruments, and regulations promote green technology. Additionally, there are studies [11,41–44] that examine the impact of environmental regulations, tools, innovations, and international environmental policy on the diffusion of green technologies in various sectors and countries, obtaining evidence that promotes green technology generation and diffusion.

Both the theoretical and empirical literature have established a link between environmental conditions and the diffusion of green technologies. Ghezzi et al. [45] investigate the Italian mobile network market's regulation, environment, strategy, and technology (REST) model. Theoretical and empirical findings add additional support to current research on the diffusion of mobile technologies by merging technology diffusion and strategy analysis. Pérez-Suárez and López-Menéndez [46] assess the performance of environmental forecasts using the sustainability-based environmental Kuznets curve (EKC) and environmental logistic curve (ELC) models. Their findings provide compelling evidence for country-specific $CO_2$ emission policies. Similarly, Hötte [8] improves the modeling of technological learning and the various shapes of diffusion curves in several dimensions by employing a macroeconomic agent-based model. The empirical and theoretical findings demonstrate the applicability of climate policies based on the type of diffusion barriers and the extent to which green technologies can be advanced. Satrovic and Ahmad [47] evaluate the existence of EKC for the Gulf Cooperation Council (GCC) countries by examining the relationships between carbon dioxide emissions per capita, urban population, per capita GDP, electric power consumption per capita, and democratic accountability index. Their findings indicate that the EKC exists for Saudi Arabia and Bahrain, but that it does not exist in the other countries, which have a U-shaped curve.

Several related studies have focused on the efficacy of environmental attitudes and awareness in promoting green technology adoption. Zeng et al. [48] investigate the dynamics of green technology diffusion and consumers' environmental attitudes, obtaining evidence that their habits and attitudes favor green technology diffusion. Hussain et al. [49], on the other hand, employ the survey approach to examine customers' attitudes and awareness toward green technology. According to the data, there is a substantial correlation between the process of acquiring green products and customer awareness. Finally, Obydenkova and Salahodjaev [50] examine the relationship between the Climate Laws, Institutions, and Measures Index (CLIMI), cognitive abilities, and democracy index in 94 countries. Their findings indicate that societies' democratic and cognitive capabilities are positively correlated with climate change policies. Other research [51–56] has empirically confirmed that consumers' habits, attitudes, and awareness processes are critical for green technology uptake.

We expand on previous research by introducing new dimensions of green technology dissemination in this study. Unlike earlier research, we concentrate on the spread of green technology rather than just the development of green technologies. We further extend the past research by introducing a broad range of factors that may influence green technology diffusion.

## 3. Methodology and Data

### 3.1. Dynamic Panel Data Model

Our data covers annual observations for N = 58 countries from 1990 to 2019, during a T = 30-year span. As a result, we must check for common panel data regression problems such as multicollinearity, autocorrelation, nonstationarity, heteroscedasticity, heterogeneity, and cross-section dependence. Unit root tests establish that the variables in our study

are stationary (see Section 4), so we use dynamic panel data methods to investigate the relationships between green technology diffusion and its determinants. Historically, the dynamic panel estimate approach has been utilized in a variety of empirical investigations on environmental issues [57–62]. We use the system generalized method moments (GMM) estimator introduced by Arellano and Bover [63] and Blundell and Bond [64] for the potential presence of endogeneity, heteroskedasticity, autocorrelation, and fixed effects. When an unobserved time-invariant country impact is missing, the ordinary least squares (OLS) technique is biased and inconsistent [65]. The Arellano–Bond GMM estimator starts by differentiating all regressors and applying the GMM method [66], resulting in differenced GMM. Furthermore, Arellano and Blundell's GMM estimation provides a method in which the first differences of instrumental variables are uncorrelated with the fixed effects, resulting in a system GMM estimator that is more efficient and consistent. The System GMM estimator is intended for situations with "small *T*, large *N*" panels, which include a small number of time periods and a large number of individuals; non-exogenous independent variables that are correlated with past and possibly current realizations of the error; fixed effects; and heteroskedasticity and autocorrelation within individuals [67].

Most studies in the literature examine the influence of $CO_2$ emissions, inequality, foreign direct investment, and democracy on green technology [10,15,68–76]. Our study extends the analysis into the effects of technological change, environmental policy, political factors, and economic performance on green technology diffusion in a large panel of world economies. As a result, this study raises new research questions about the effects of technological, political, social, environmental, and economic elements on the diffusion of green technology. Also, our study examines these effects using a dynamic system GMM model rather than static models such as the fixed effects model.

The most general form of the model in our study is as follows:

$$\text{ETD}_{i,t} = f(\text{DA}_{i,t}, \text{EDU}_{i,t}, \text{FDIR}_{i,t}, \text{SOC}_{i,t}, \text{TOP1}_{i,t}, \text{CO}_{2,i,t}, \\ \text{EPI}_{i,t}, \text{GDP}_{i,t}, \text{GTD}_{i,t}, \text{TAI}_{i,t}) + \varepsilon_{i,t} \tag{1}$$

where EDT represents environmental or green technology diffusion which is a function of ten variables: general technology diffusion (GTD), technology achievement index (TAI), per capita gross domestic product measured in US dollars (GDP), income inequality measured as share of the top 1% of highest-income households (TOP1), education index (EDU), socio-economic conditions (SOC), democratic accountability (DA), foreign direct investment as percent of gross domestic product (FDIR), $CO_2$ emission (metric tons per capita), and EPI environmental performance index (EPI). In equation $t = 1, 2, \ldots, T$ denotes time in years and $i = 1, 2, \ldots, N$ denotes countries. All variables are in natural logarithms.

We assume that the relationship between the dependent variable and the independent variables is linear. Thus, the explicit form of the dynamic panel model can be written as follows:

$$\ln(\text{ETD}_{i,t}) = \alpha_0 + \sum_{j=1}^{P} \rho_i \ln(\text{ETD}_{i,t-j}) + \beta_1 \ln(\text{DA}_{i,t}) + \beta_2 \ln(\text{EDU}_{i,t}) + \beta_3 \ln(\text{FDIR}_{i,t}) + \beta_4 \ln(\text{SOC}_{i,t}) \\ + \beta_5 \ln(\text{TOP1}_{i,t}) + \beta_6 \ln(\text{CO}_{2,i,t}) + \beta_7 \ln(\text{EPI}_{i,t}) + \beta_8 \ln(\text{GDP}_{i,t}) + \beta_9 \ln(\text{GTD}_{i,t}) \\ + \beta_{10} \ln(\text{TAI}_{i,t}) + \eta_i + v_{i,t} \tag{2}$$

where ln denotes natural logarithm, p is the autocorrelation order, and $\eta_i + v_{i,t}$ is the usual error components decomposition of the error term. As usual, we assume that $\eta_i$ and $v_{i,t}$ are independently distributed across i and have the familiar error component's structure, that is, $E(\eta_i) = 0$, $E(v_{i,t})$, and $E(v_{i,t}\eta_i) = 0$. Moreover, the errors $v_{i,t}$ are not autocorrelated, i.e., $E(v_{i,t}v_{i,s}) = 0$ for $t \neq s$.

We use the first to second lags of all variables included in the regression as GMM-style instruments in our implementation of the Arellano-Bond system GMM model. To ensure that all relevant variables are included as instruments while avoiding biasing our parameters, we include one instrument for each variable and lag distance rather than one

instrument for each variable, time period, and lag distance. This was done because when the number of instruments included increases in proportion to the number of data, the parameter estimates become biased toward feasible generalized least squares estimates [64].

The validity of all instruments in the regression model can be evaluated using the Saragn-Hansen *J*-test of over-identifying restrictions. We also employ the serial correlation tests of Arellano and Bond [77]. In the presence of a lagged endogenous variable as independent variable in the regression model, the two-step system GMM estimator allows for an exact degree of endogeneity. As a result, we estimate the empirical linkages between green technology diffusion and chosen variables using Arellano and Bover's (1995) and Blundell and Bond's (1998) two-step system GMM estimator.

Due to the significant correlation between the variables EDU, TAI, EPI, GDP, $CO_2$, and GDT, we are unable to estimate the whole model in Equation (2). Because of the nature of the economy or because a variable is already included in an index variable, there is a high correlation among these variables. EDU, for example, is a sub-component of the TAI variable. As a result, in Equation (2), we impose various constraints and estimate 11 alternative forms of the generic model. These models, along with the restrictions they imply on equation two, are given in Table 1. When we exclude a variable or a group of variables, we consider whether one of the remaining index variables in the model includes the excluded variable as a component, whether the excluded variable may measure the same concept that another variable in the model already measures, or whether there is a high correlation between some variables that results in a serious multicollinearity problem.

**Table 1.** Model specifications.

| Model Name | Exclusion Restriction | Excluded Variables |
|:---:|:---:|:---:|
| Model 1 | $\beta_6 = \beta_7 = \beta_8 = \beta_9 = \beta_{10} = 0$ | $CO_2$, EPI, GDP, GTD, TAI |
| Model 2 | $\beta_2 = \beta_7 = \beta_8 = \beta_9 = \beta_{10} = 0$ | EDU, EPI, GDP, GTD, TAI |
| Model 3 | $\beta_6 = \beta_8 = \beta_9 = \beta_{10} = 0$ | $CO_2$, GDP, GTD, TAI |
| Model 4 | $\beta_2 = \beta_6 = \beta_7 = \beta_9 = \beta_{10} = 0$ | EDU, $CO_2$, EPI, GTD, TAI |
| Model 5 | $\beta_6 = \beta_7 = \beta_8 = \beta_{10} = 0$ | $CO_2$, EPI, GDP, TAI |
| Model 6 | $\beta_2 = \beta_6 = \beta_7 = \beta_8 = \beta_9 = 0$ | EDU, $CO_2$, EPI, GDP, GTD |
| Model 7 | $\beta_8 = \beta_9 = \beta_{10} = 0$ | GDP, GTD, TAI |
| Model 8 | $\beta_2 = \beta_6 = \beta_8 = \beta_9 = 0$ | EDU, $CO_2$, GDP, GTD |
| Model 9 | $\beta_2 = \beta_9 = \beta_{10} = 0$ | EDU, GTD, TAI |
| Model 10 | $\beta_2 = \beta_8 = \beta_{10} = 0$ | EDU, GDP, TAI |
| Model 11 | $\beta_2 = \beta_6 = \beta_{10} = 0$ | EDU, $CO_2$, TAI |
| | **Implied relationship** | |
| Model 1 | ETD = f(DA, EDU, FDIR, SOC, TOP1) + $\varepsilon$ | |
| Model 2 | ETD = f(DA, FDIR, SOC, TOP1, $CO_2$) + $\varepsilon$ | |
| Model 3 | ETD = f(DA, EDU, FDIR, SOC, TOP1, EPI) + $\varepsilon$ | |
| Model 4 | ETD = f(DA, FDIR, SOC, TOP1, GDP) + $\varepsilon$ | |
| Model 5 | ETD = f(DA, EDU, FDIR, SOC, TOP1, GTD) + $\varepsilon$ | |
| Model 6 | ETD = f(DA, FDIR, SOC, TOP1, TAI) + $\varepsilon$ | |
| Model 7 | ETD = f(DA, EDU, FDIR, SOC, TOP1, $CO_2$, EPI) + $\varepsilon$ | |
| Model 8 | ETD = f(DA, FDIR, SOC, TOP1, EPI, TAI) + $\varepsilon$ | |
| Model 9 | ETD = f(DA, FDIR, SOC, TOP1, $CO_2$, EPI, GDP) + $\varepsilon$ | |
| Model 10 | ETD = f(DA, FDIR, SOC, TOP1, $CO_2$, EPI, GTD) + $\varepsilon$ | |
| Model 11 | ETD = f(DA, FDIR, SOC, TOP1, EPI, GDP, GTD) + $\varepsilon$ | |

We use four essential diagnostics after estimating each model utilizing the two-step system GMM method. The first is the Sargan-Hansen *J*-test of overidentifying restrictions [66,78], which is used to assess the validity of an instrument. Second, the first-order [AR(1)] and second-order [AR(2)] autocorrelation tests are employed to see if enough lags are included to account for autocorrelation.

### 3.2. Data

The study uses annual panel data over the period 1990–2019 for 58 countries, which are listed in Appendix A. These countries are selected based on data availability and include developing and developed countries. We construct the green technology diffusion and general technology diffusion using the World Intellectual Property Organization's (WIPO) patent dataset. This WIPO dataset contains patent indicators and statistics that are useful for assessing environmental innovations. This WIPO dataset enables the assessment of countries' and firms' innovation performance, as well as the development of governmental environmental and innovation policies. The average of the green technology diffusion and general technology diffusion variables across 58 countries for the period 1990–2019 is depicted in Figure 1. As illustrated in Figure 1, both general technology diffusion and green technology dissemination increased in tandem until 2010. However, since 2010, green technology diffusion has increased at a faster rate than general technology diffusion.

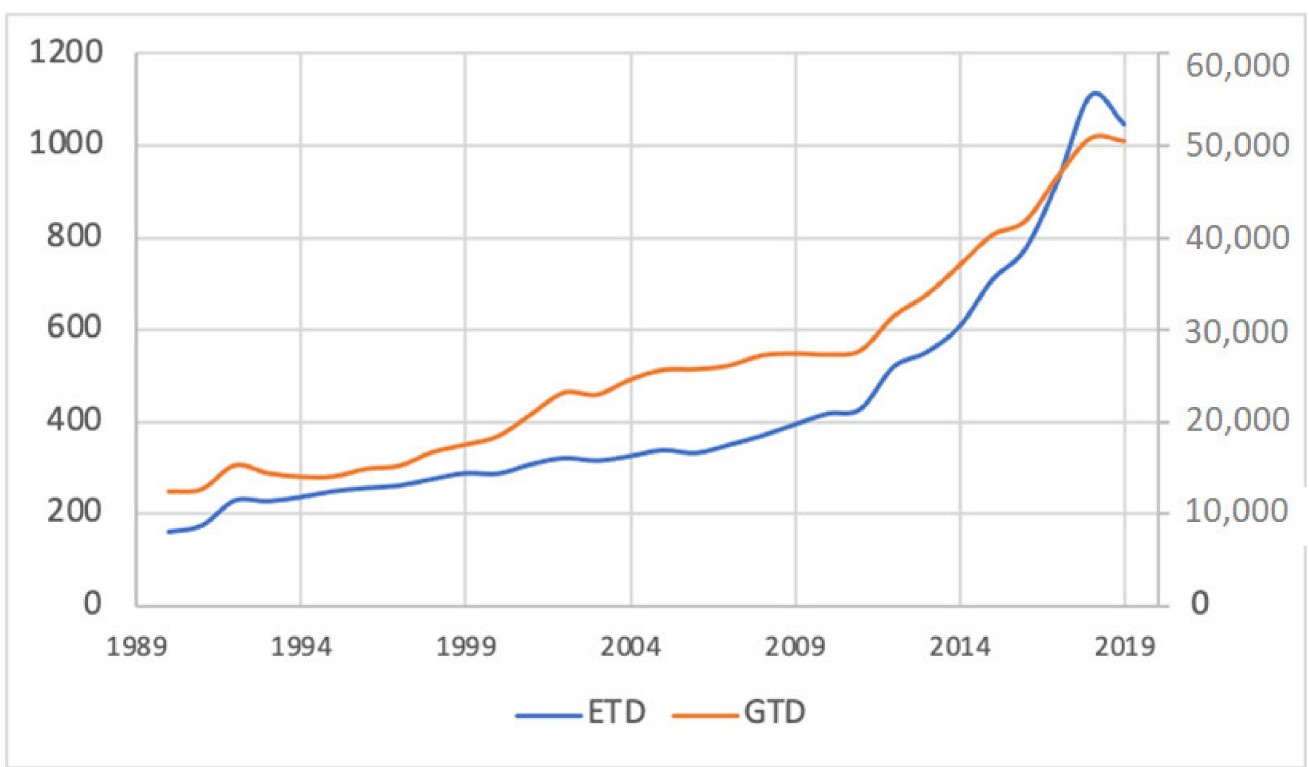

**Figure 1.** A time series plot of the average green technology diffusion and the average general technology diffusion.

GDP per capita figures, which are purchasing power parity (PPP)-adjusted and in US dollars, as well as per capita $CO_2$ emissions in metric tons, are sourced from the World Bank's World Development Indicators (WDI) database. Additionally, we collect foreign direct investment from the WDI, which we convert to a proportion of GDP. The income share of the top 1% of the wealthiest households is derived from World Inequality. The factors pertaining to democratic accountability and socioeconomic situations are drawn from the Political Risk Services (PRS) Group database. Both variables are indexed from 1 to 10, with a higher index value indicating better outcomes.

Education index is extracted from the Human Development Report of the United Nations. The education index calculation is based on an average of expected years of schooling (of children) and means years of schooling (of adults) which is normalized by scaling with the corresponding maxima. The environmental performance index is retrieved from the Socioeconomic Data and Applications Center. The environmental performance

index has a data cluster based on 25 indicators and 10 policy categories. These policies are related to the environmental burden of disease, water (effects on human health), air pollution (effects on human health), water resources, biodiversity and habitat, forestry, fisheries, agriculture, and climate change [79].

Finally, technological advancements are critical for economic growth and sustainable development. Its objective is to advance sustainability through countries' capabilities and usage of technological and environmental investments. We quantify technological achievement by developing a technology achievement index using data gathered from multiple data sources, such as databases maintained by various organizations. We generate the TAI using the method developed by Desai et al. [17]. The TAI has four dimensions and sub-indicators: human skill development, new technology creation, old technology diffusion, and new technology diffusion. Gross enrolment ratios at all levels (excluding pre-primary) and gross enrollment ratios at the tertiary level are used to determine human skill development. Patents granted to residents and royalties and license fees collected (in US dollars per person) are used to quantify the development of new technology. Access to energy (kWh per capita) and telephone mainlines, as well as cellular users, contribute to the diffusion of old technology (per 1000 people). The diffusion of new technology is measured in terms of internet users (per 1000 people) and high-tech exports (percent of manufactured exports). Figure 2 depicts the average TAI for 58 countries from 1990 to 2019. After 2005, we observe a slowing of TAI growth, culminating in near-stagnation in 2015.

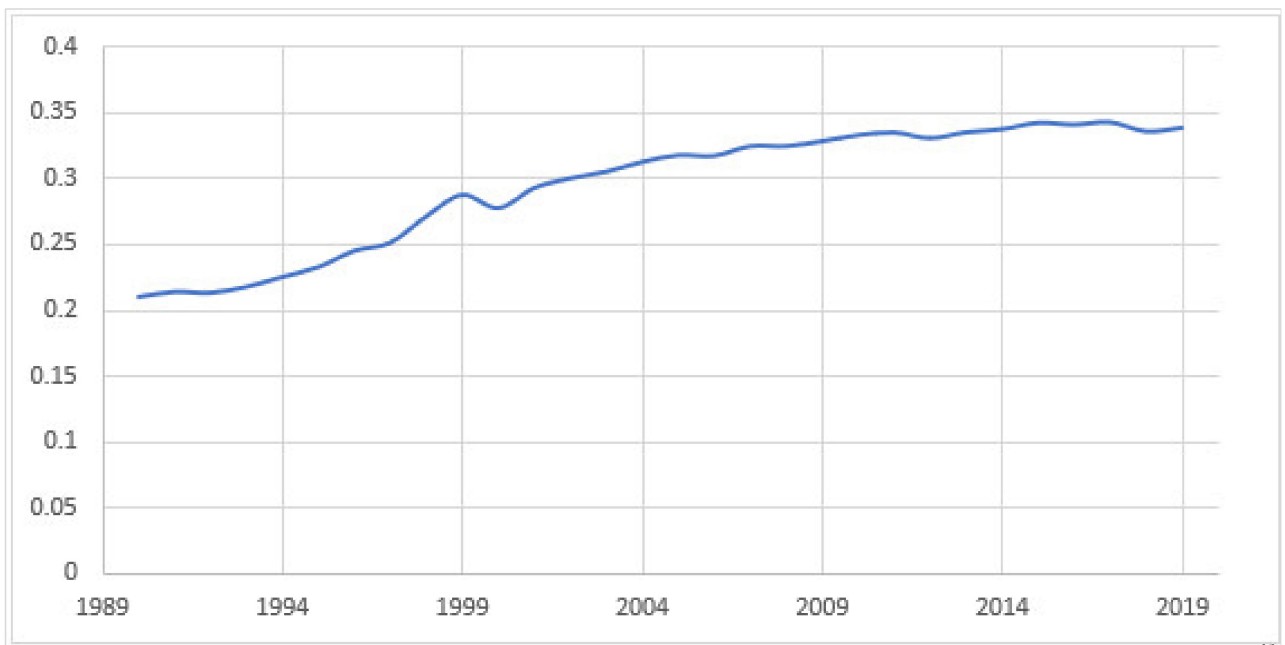

**Figure 2.** The average technology achievement index for the whole period.

Table 2 summarizes the statistics for all panel data variables, including the number of observations (n), the mean, the standard deviation, and the minimum and maximum values. There are 1740 observations in our balanced data sets. The second panel of Table 2 contains descriptive statistics for level logarithms, where the $\ln(1 + x)$ transformation is utilized in circumstances where direct logarithmic transformation is not possible. However, when untransformed data is used, our results are qualitatively same. The second panel's first column contains the variable names for logarithmic levels that match to the names in the first panel.

**Table 2.** Descriptive statistics.

| Variable | *n* | Mean | Std. Dev. | Min | Max |
|---|---|---|---|---|---|
| | | *Levels* | | | |
| ETD | 1740 | 425.147 | 2126.166 | 0.000 | 44,195.000 |
| DA | 1740 | 4.706 | 1.641 | 0.042 | 6.000 |
| EDU | 1740 | 0.726 | 0.135 | 0.254 | 0.943 |
| FDIR | 1740 | 0.010 | 0.026 | −0.062 | 0.272 |
| SOC | 1740 | 6.595 | 2.335 | 0.167 | 11.000 |
| TOP1 | 1740 | 0.134 | 0.054 | 0.037 | 0.378 |
| $CO_2$ | 1740 | 7.378 | 4.564 | 0.675 | 27.431 |
| EPI | 1740 | 59.337 | 11.559 | 29.090 | 90.680 |
| GDP | 1740 | 23,488.320 | 17,159.730 | 982.501 | 120,962.200 |
| GTD | 1740 | 25,899.140 | 98,867.840 | 0.000 | 1,423,528.000 |
| TAI | 1740 | 0.295 | 0.121 | 0.021 | 0.820 |
| | | *Log levels* | | | |
| LETD | 1740 | 3.399 | 2.229 | 0.000 | 10.696 |
| LGTD | 1740 | 7.203 | 2.585 | 0.000 | 14.169 |
| LEPI | 1740 | 4.063 | 0.203 | 3.370 | 4.507 |
| $LCO_2$ | 1740 | 1.773 | 0.737 | −0.393 | 3.312 |
| LGDP | 1740 | 9.777 | 0.816 | 6.890 | 11.703 |
| LTAI | 1740 | −1.318 | 0.472 | −3.863 | −0.199 |
| LTOP1 | 1740 | −2.081 | 0.383 | −3.310 | −0.973 |
| LEDU | 1740 | −0.340 | 0.210 | −1.370 | −0.059 |
| LDA | 1740 | 1.416 | 0.664 | −3.178 | 1.792 |
| LSOC | 1740 | 1.762 | 0.652 | −1.792 | 2.398 |
| LFDIR | 1740 | 0.926 | 2.449 | −6.446 | 24.053 |

Note: *n* is the number of observations while Std. dev. denotes standard deviation.

The Pearson correlation coefficient estimates for all variables are shown in Table 3. Correlation estimates indicate whether there is a high degree of correlation between the model's variables, which may aid in identifying possible multicollinearity. The correlation coefficients between LETD and LTOP1 are found to be negative. Other variables, on the other hand, are positively correlated with the diffusion of green technologies. Additionally, some pairs of variables have a high and positive correlation coefficient. LETD and LGTD, LTAI and LEDU, LTAI and LGDP, LGDP and LEDU, LGDP and LEPI, LGDP and $LCO_2$, and $LCO_2$ and LEDU are some of these pairs.

**Table 3.** The Pearson correlation coefficient estimates.

| Variable | LETD | LDA | LEDU | LFDIR | LSOC | LTOP1 | $LCO_2$ | LEPI | LGDP | LGTD | LTAI |
|---|---|---|---|---|---|---|---|---|---|---|---|
| LETD | 1.00 | | | | | | | | | | |
| LDA | **0.28** | 1.00 | | | | | | | | | |
| LEDU | **0.42** | **0.33** | 1.00 | | | | | | | | |
| LFDIR | **0.39** | −0.09 | −0.12 | 1.00 | | | | | | | |
| LSOC | **0.37** | **0.67** | **0.31** | 0.11 | 1.00 | | | | | | |
| LTOP1 | −0.16 | −0.02 | −0.36 | 0.10 | 0.10 | 1.00 | | | | | |
| $LCO_2$ | **0.42** | 0.04 | **0.59** | −0.04 | **0.19** | −0.40 | 1.00 | | | | |
| LEPI | **0.30** | **0.32** | **0.62** | −0.18 | **0.32** | −0.28 | 0.25 | 1.00 | | | |
| LGDP | **0.44** | **0.33** | **0.76** | −0.12 | **0.53** | −0.21 | **0.64** | **0.69** | 1.00 | | |
| LGTD | **0.96** | **0.34** | **0.46** | **0.37** | **0.44** | −0.16 | **0.41** | **0.32** | **0.51** | 1.00 | |
| LTAI | **0.59** | **0.31** | **0.85** | 0.12 | **0.32** | −0.34 | **0.57** | **0.52** | **0.70** | **0.64** | 1.00 |

Note: Boldface indicates significance at 1% level.

We give the mean of variables by country in Table 4. For each country, the mean is calculated using the data from 1990 to 2019. China, Japan, and the United States all have the greatest mean for green technology diffusion, whereas Japan, the US, and China all have the highest mean for general technology diffusion. Similarly, the US, Japan, Canada,

and Sweden have the highest mean for the technology achievement index. Switzerland, Norway, and Luxembourg have the highest mean in the environmental performance index.

**Table 4.** The means of variables by country for the period 1990–2019.

| Country | ETD | DA | EDU | FDIR | SOC | TOP1 | CO$_2$ | EPI | GDP | GTD | TAI |
|---|---|---|---|---|---|---|---|---|---|---|---|
| Argentina | 7.37 | 4.45 | 0.75 | 0.00 | 5.40 | 0.19 | 4.12 | 55.32 | 14,713.19 | 473.13 | 0.25 |
| Australia | 148.33 | 6.00 | 0.90 | 0.01 | 8.51 | 0.10 | 16.52 | 61.79 | 33,124.07 | 8827.03 | 0.34 |
| Austria | 137.60 | 5.64 | 0.77 | 0.00 | 8.62 | 0.11 | 7.76 | 70.99 | 36,194.82 | 7582.67 | 0.34 |
| Belarus | 10.80 | 1.49 | 0.73 | 0.00 | 3.72 | 0.10 | 6.40 | 55.07 | 10,805.19 | 564.60 | 0.29 |
| Belgium | 112.70 | 5.83 | 0.84 | 0.01 | 7.74 | 0.08 | 10.09 | 61.60 | 33,792.42 | 7425.20 | 0.35 |
| Brazil | 85.70 | 4.33 | 0.59 | 0.03 | 5.93 | 0.26 | 1.94 | 58.23 | 11,232.41 | 4266.63 | 0.17 |
| Bulgaria | 9.13 | 5.04 | 0.72 | 0.00 | 4.75 | 0.12 | 6.27 | 56.00 | 11,941.18 | 435.77 | 0.26 |
| Canada | 285.10 | 5.97 | 0.84 | 0.01 | 8.35 | 0.13 | 16.09 | 60.85 | 34,474.07 | 16,496.73 | 0.43 |
| China | 7684.33 | 1.58 | 0.53 | 0.17 | 6.87 | 0.12 | 4.66 | 43.93 | 6597.00 | 289,318.70 | 0.30 |
| Colombia | 6.00 | 4.00 | 0.57 | 0.01 | 4.45 | 0.22 | 1.69 | 60.64 | 9189.29 | 201.37 | 0.18 |
| Croatia | 6.17 | 3.76 | 0.68 | 0.00 | 3.79 | 0.08 | 4.39 | 63.62 | 16,023.12 | 300.53 | 0.25 |
| Cyprus | 3.70 | 5.59 | 0.72 | 0.00 | 8.63 | 0.11 | 6.66 | 60.06 | 26,159.24 | 242.17 | 0.25 |
| Czechia | 40.03 | 5.26 | 0.78 | 0.00 | 7.13 | 0.09 | 11.30 | 65.34 | 23,265.93 | 1216.87 | 0.32 |
| Denmark | 121.20 | 6.00 | 0.85 | 0.00 | 8.78 | 0.11 | 8.95 | 66.31 | 35,602.46 | 6842.27 | 0.38 |
| Egypt | 2.97 | 2.51 | 0.50 | 0.00 | 5.33 | 0.18 | 2.07 | 53.54 | 7543.04 | 99.13 | 0.11 |
| Estonia | 5.70 | 3.96 | 0.81 | 0.00 | 5.16 | 0.13 | 12.81 | 60.07 | 17,579.90 | 197.17 | 0.33 |
| Finland | 209.23 | 6.00 | 0.84 | 0.00 | 8.31 | 0.09 | 10.38 | 66.79 | 32,149.65 | 10,016.17 | 0.41 |
| France | 792.47 | 5.79 | 0.76 | 0.01 | 7.58 | 0.10 | 5.63 | 68.59 | 30,908.85 | 52,272.27 | 0.38 |
| Germany | 2186.00 | 5.64 | 0.86 | 0.02 | 7.80 | 0.11 | 9.87 | 69.76 | 34,398.31 | 132,895.40 | 0.40 |
| Greece | 16.87 | 5.57 | 0.73 | 0.00 | 6.12 | 0.11 | 7.54 | 61.65 | 23,045.17 | 696.23 | 0.26 |
| Hungary | 41.67 | 5.45 | 0.76 | 0.01 | 6.52 | 0.10 | 5.38 | 58.22 | 17,559.98 | 1490.33 | 0.26 |
| Iceland | 0.63 | 6.00 | 0.82 | 0.00 | 8.11 | 0.09 | 6.97 | 68.23 | 36,525.55 | 199.67 | 0.40 |
| India | 36.07 | 5.36 | 0.43 | 0.04 | 5.06 | 0.18 | 1.21 | 36.09 | 3316.96 | 3494.80 | 0.18 |
| Ireland | 28.47 | 5.96 | 0.82 | 0.01 | 8.59 | 0.11 | 9.21 | 61.08 | 40,124.76 | 2789.33 | 0.32 |
| Israel | 76.70 | 5.70 | 0.83 | 0.00 | 7.26 | 0.16 | 8.67 | 58.28 | 26,739.77 | 7097.27 | 0.33 |
| Italy | 262.20 | 4.95 | 0.73 | 0.01 | 7.52 | 0.08 | 7.02 | 68.64 | 30,524.54 | 19,597.33 | 0.30 |
| Japan | 6129.67 | 5.24 | 0.78 | 0.00 | 8.09 | 0.11 | 9.33 | 64.44 | 30,999.44 | 436,595.90 | 0.49 |
| Kazakhstan | 5.87 | 1.18 | 0.74 | 0.00 | 4.86 | 0.13 | 12.53 | 39.14 | 14,608.92 | 236.77 | 0.23 |
| Latvia | 3.33 | 3.63 | 0.77 | 0.00 | 4.45 | 0.09 | 3.70 | 67.51 | 14,774.18 | 205.73 | 0.28 |
| Lithuania | 5.37 | 3.91 | 0.79 | 0.00 | 4.79 | 0.10 | 4.48 | 65.05 | 16,602.91 | 145.40 | 0.30 |
| Luxembourg | 27.90 | 5.89 | 0.73 | 0.00 | 9.46 | 0.13 | 20.86 | 71.88 | 70,381.46 | 1702.60 | 0.24 |
| Mexico | 26.23 | 5.04 | 0.59 | 0.02 | 7.06 | 0.24 | 4.05 | 49.09 | 13,356.05 | 1009.77 | 0.21 |
| Moldova | 6.73 | 3.16 | 0.66 | 0.00 | 2.86 | 0.09 | 2.57 | 46.81 | 5784.08 | 240.33 | 0.28 |
| Morocco | 4.67 | 3.73 | 0.40 | 0.00 | 5.59 | 0.16 | 1.42 | 47.92 | 4936.03 | 125.20 | 0.06 |
| Netherlands | 358.53 | 6.00 | 0.85 | 0.03 | 8.85 | 0.07 | 10.85 | 67.52 | 37,661.82 | 24,388.67 | 0.38 |
| New Zealand | 17.90 | 6.00 | 0.88 | 0.00 | 8.37 | 0.10 | 7.74 | 65.75 | 26,966.66 | 1557.67 | 0.32 |
| Norway | 93.13 | 5.93 | 0.88 | 0.00 | 9.44 | 0.11 | 8.78 | 71.50 | 44,964.18 | 3861.90 | 0.42 |
| Peru | 1.97 | 3.93 | 0.64 | 0.00 | 5.28 | 0.28 | 1.34 | 50.87 | 7504.84 | 52.70 | 0.23 |
| Philippines | 2.37 | 4.78 | 0.60 | 0.01 | 5.13 | 0.19 | 0.91 | 53.74 | 4833.73 | 108.37 | 0.12 |
| Poland | 123.03 | 5.44 | 0.79 | 0.01 | 5.94 | 0.12 | 8.30 | 64.48 | 16,262.97 | 3436.93 | 0.31 |
| Portugal | 10.63 | 5.61 | 0.68 | 0.00 | 7.23 | 0.11 | 5.15 | 58.83 | 22,474.35 | 544.20 | 0.26 |
| Romania | 22.47 | 5.39 | 0.70 | 0.00 | 4.86 | 0.12 | 4.55 | 48.84 | 12,840.81 | 1066.97 | 0.24 |
| Russia | 588.63 | 2.80 | 0.74 | 0.01 | 4.94 | 0.20 | 12.39 | 51.52 | 14,644.80 | 25,857.43 | 0.31 |
| Saudi Arabia | 21.53 | 1.02 | 0.63 | 0.00 | 6.81 | 0.18 | 16.00 | 53.22 | 44,658.53 | 580.50 | 0.21 |
| Serbia | 2.10 | 4.18 | 0.70 | 0.00 | 3.51 | 0.11 | 7.46 | 50.43 | 9897.00 | 105.30 | 0.21 |
| Singapore | 27.67 | 2.30 | 0.70 | 0.00 | 8.90 | 0.13 | 10.65 | 59.09 | 58,887.87 | 2731.97 | 0.30 |
| Slovakia | 11.43 | 5.45 | 0.76 | 0.00 | 6.42 | 0.07 | 6.99 | 65.47 | 18,183.64 | 299.57 | 0.31 |
| Slovenia | 9.33 | 3.64 | 0.82 | 0.00 | 4.73 | 0.07 | 7.11 | 62.88 | 23,343.57 | 571.03 | 0.35 |
| South Africa | 41.97 | 4.64 | 0.66 | 0.00 | 5.04 | 0.16 | 8.72 | 38.77 | 9615.10 | 1757.37 | 0.19 |
| Spain | 108.30 | 5.79 | 0.74 | 0.01 | 6.71 | 0.12 | 6.31 | 62.63 | 26,520.22 | 6609.40 | 0.32 |
| Sweden | 266.83 | 6.00 | 0.85 | 0.00 | 8.11 | 0.10 | 5.44 | 70.46 | 35,556.45 | 18,433.20 | 0.43 |
| Switzerland | 262.83 | 6.00 | 0.82 | 0.00 | 9.37 | 0.11 | 5.29 | 78.91 | 45,063.12 | 27,650.27 | 0.39 |
| Thailand | 2.87 | 3.79 | 0.55 | 0.01 | 7.07 | 0.22 | 3.24 | 56.39 | 10,683.14 | 174.20 | 0.21 |
| Turkey | 15.80 | 4.20 | 0.55 | 0.00 | 5.20 | 0.20 | 3.66 | 45.47 | 15,307.26 | 1656.87 | 0.19 |
| UK | 550.17 | 5.91 | 0.84 | 0.03 | 8.73 | 0.17 | 8.24 | 68.15 | 31,502.90 | 37,067.67 | 0.42 |
| US | 3588.67 | 5.92 | 0.87 | 0.05 | 8.53 | 0.12 | 18.25 | 59.20 | 42,436.14 | 328,214.90 | 0.73 |
| Uruguay | 0.93 | 4.75 | 0.68 | 0.00 | 5.84 | 0.19 | 1.82 | 58.38 | 13,383.80 | 56.40 | 0.19 |
| Venezuela | 2.53 | 3.87 | 0.59 | 0.00 | 4.31 | 0.21 | 6.21 | 56.55 | 14,155.73 | 66.00 | 0.17 |
| All countries | 425.15 | 4.71 | 0.73 | 0.01 | 6.60 | 0.13 | 7.38 | 59.34 | 23,488.32 | 25,899.14 | 0.29 |

## 4. Empirical Results and Discussion

### 4.1. Empirical Results

We begin by reporting the results of essential preliminary tests, such as cross-sectional dependency, homogeneity, and unit root tests. Cross-sectional dependence is critical when examining the relationships between variables in panel data models. Due to the prospect of countries becoming dependent on one another as a result of numerous economic ties and the effect of shared causes, ignoring spillover effects might result in incorrect inference and misspecification issues. Thus, we begin by examining cross-sectional dependence. Additionally, slope homogeneity tests are included. As shown in Table 5, we report five cross-sectional dependence tests: the Lagrange multiplier (LM) test developed by Breusch and Pagan [80], the adjusted Lagrange multiplier (LM-adj) test developed by Pesaran et al. [81], the cross-sectional dependence (CD) test developed by Pesaran [82,83], the LM cross-sectional dependence ($CD_{LM}$) test developed by Pesaran [82,83], and the adjusted $CD_{LM}$ ($CD_{LM}$-adj) test developed by Baltagi et al. [84]. Another critical preliminary test in panel estimation is the model coefficients' country-specific heterogeneity. We employ two alternative tests to determine slope homogeneity. The first test is Pesaran and Yamagata's [85] truncated slope homogeneity ($\widetilde{\Delta}_{HAC}$) test with Blomquist and Westerlund's [86] heteroskedasticity and autocorrelation consistent covariance (HAC) adjustment. The second test is an adjusted version of the $\widetilde{\Delta}_{HAC}$ test for small samples, designated by $\widetilde{\Delta}_{adj, HAC}$. Each test is constructed using a pooled ordinary least squares regression with six different model specifications. LETD is the dependent variable in each model. Each of the Models 1 to 5 include the variables LDA, LEDU, LFDIR, LSOC, and LTOP1, but each also includes one of the variables $LCO_2$, LEPI, LGDP, LGTD, and LTAI as independent variables, in the provided order. Model 6 augments the independent variables in Model 1 with the variables $LCO_2$, LEPI, LGDP, LGTD, and LTAI.

**Table 5.** The cross-sectional dependence and slope homogeneity tests.

| Test | Statistic | *p*-Value | Statistic | *p*-Value |
|---|---|---|---|---|
| | *Test in Model 1* | | *Test in Model 4* | |
| LM | 2183.630 *** | 0.000 | 1943.164 *** | 0.000 |
| LM-adj | 5.747 *** | 0.000 | 1.300 | 0.194 |
| $CD_{LM}$ | 9.229 *** | 0.000 | 5.047 *** | 0.000 |
| $CD_{LM}$-adj | 8.229 *** | 0.000 | 4.047 *** | 0.000 |
| CD | 20.234 *** | 0.000 | 1.851 * | 0.064 |
| $\widetilde{\Delta}_{HAC}$ | 1.318 | 0.187 | 1.522 | 0.128 |
| $\widetilde{\Delta}_{adj, HAC}$ | 1.566 | 0.117 | 1.808 * | 0.071 |
| | *Test in Model 2* | | *Test in Model 5* | |
| LM | 2061.656 *** | 0.000 | 2098.710 *** | 0.000 |
| LM-adj | 3.107 *** | 0.002 | 3.911 *** | 0.000 |
| $CD_{LM}$ | 7.107 *** | 0.000 | 7.752 *** | 0.000 |
| $CD_{LM}$-adj | 6.107 *** | 0.000 | 6.752 *** | 0.000 |
| CD | 23.936 *** | 0.000 | 13.215 *** | 0.000 |
| $\widetilde{\Delta}_{HAC}$ | 1.086 | 0.277 | 0.896 | 0.370 |
| $\widetilde{\Delta}_{adj, HAC}$ | 1.291 | 0.197 | 1.065 | 0.287 |
| | *Test in Model 3* | | *Test in Model 6* | |
| LM | 2087.873 *** | 0.000 | 1993.615 *** | 0.000 |
| LM-adj | 3.613 *** | 0.000 | 0.427 | 0.669 |
| $CD_{LM}$ | 7.563 *** | 0.000 | 5.924 *** | 0.000 |
| $CD_{LM}$-adj | 6.563 *** | 0.000 | 4.924 *** | 0.000 |
| CD | 20.456 *** | 0.000 | 0.093 | 0.926 |
| $\widetilde{\Delta}_{HAC}$ | 1.094 | 0.274 | –0.565 | 0.572 |
| $\widetilde{\Delta}_{adj, HAC}$ | 1.300 | 0.194 | –0.931 | 0.352 |

Note: *, **, and *** denote significance at 10%, 5%, and 1% levels, respectively.

The cross-sectional dependence tests shown in Table 5 indicate that, with just three exceptions, the null hypothesis of no cross-sectional dependence is rejected for all countries in all models at all significant levels. Thus, the results reveal the existence of cross-sectional dependence among countries, implying that, as a result of the 58 countries' high degree of integration, the shock that originated in one of these 58 countries appears to have propagated to the others. Another significant point is that the unit root tests should be chosen depending on the cross-sectional dependence test results. Due to the existence of cross-sectional dependence in the data we employ, the second-generation unit root test should be employed to ensure that the tests are efficient.

Additionally, Table 5 includes the results of slope homogeneity testing. At all levels of significance, these tests do not reject the null hypothesis of slope homogeneity across countries. As a result, the slope is unlikely to vary by country, and the influence of independent variables on green technology diffusion appears to be homogeneous across 58 countries.

The panel data method that should be utilized is determined by the data's stationarity property. Thus, we conduct unit roots tests prior to estimating empirical relationships. We employ second-generation unit root tests due to the existence of cross-sections in the data. Among the second-generation unit root tests that allow for cross-sectional dependency, we use the cross-section augmented Im-Peseran-Shin (CIPS) test of Peseran [87], the modified CIPS (M-CIPS) tests of Westerlund and Hosseinkouchack [88], the panel analysis of non-stationarity in idiosyncratic and common components (PANIC) based panel unit root test (PANIC-$Z_{\hat{e}}$) of Westerlund and Larsson [89], and the bias adjusted version of the PANIC-$Z_{\hat{e}}$ test (PANIC-$Z_{\hat{e}}^{+}$). The results of the unit root tests are given in Table 6. The results in Table 6 demonstrate that all tests substantially reject the unit root null hypothesis with constant and constant plus trends specifications at the 1% significance level, with the exception of $LCO_2$, for which the PANIC-based tests do not strongly reject the unit root null. Given that the CIPS and M-CIPS tests agree in all cases, we infer that all series are stationary and can be estimated using the stationary GMM.

**Table 6.** The panel unit root tests.

| Variable | Tests with a Constant | | | | Tests with a Constant and Trend | | | |
|---|---|---|---|---|---|---|---|---|
| | CIPS | M-CIPS | PANIC- $Z_{\hat{e}}$ | PANIC- $Z_{\hat{e}}^{+}$ | CIPS | M-CIPS | PANIC- $Z_{\hat{e}}$ | PANIC- $Z_{\hat{e}}^{+}$ |
| LETD | −3.242 ** | −20.948 ** | 0.351 | −6.539 ** | −3.754 ** | −22.925 ** | −11.739 ** | −9.423 ** |
| LDA | −2.678 ** | −13.952 ** | −2.933 ** | −0.098 | −3.189 ** | −18.324 ** | −11.557 ** | −1.699 * |
| LEDU | −2.524 ** | −12.965 ** | −8.521 ** | −10.696 ** | −2.964 ** | −16.216 ** | −12.564 ** | −15.668 ** |
| LFDIR | −3.357 ** | −22.794 ** | −9.841 ** | −11.148 ** | −3.696 ** | −25.112 ** | −13.444 ** | −8.661 ** |
| LSOC | −3.064 ** | −18.529 ** | −4.000 ** | −7.062 ** | −3.359 ** | −19.583 ** | −12.418 ** | −10.851 ** |
| LTOP1 | −2.536 ** | −12.654 ** | 0.229 | 20.017 | −3.096 ** | −17.190 ** | −10.183 ** | −10.919 ** |
| $LCO_2$ | −1.225 | −7.527 ** | −0.150 | 18.714 | −1.929 | −9.050 ** | −10.333 ** | 9.327 |
| LEPI | −2.604 ** | −15.518 ** | −3.075 ** | −4.206 ** | −2.939 ** | −13.626 ** | −11.114 ** | −89.361 ** |
| LGDP | −2.243 ** | −11.970 ** | −6.008 ** | −3.786 ** | −2.754 ** | −17.182 ** | −10.324 ** | −20.810 ** |
| LGTD | −2.612 ** | −14.404 ** | 9.164 | 8.626 | −3.012 ** | −15.136 ** | −8.286 ** | −10.769 ** |
| LTAI | −2.327 ** | −11.195 ** | 4.408 | 7.540 | −2.763 ** | −14.096 ** | −7.834 ** | −36.558 ** |

Note: ** and * denote significance at the 1% and 5% significance level, respectively. The null hypothesis for all tests is the existence of a unit root.

Since all series are stationary, and the concerns are of endogeneity, multicollinearity, autocorrelation, the system GMM method is used. We provide the system GMM estimates in Table 7, together with their standard errors, which are given in parentheses below the parameter estimates. Baltagi [90] points out that the system GMM estimator has the most desirable properties in the presence of endogenous regressors for stationary dynamic panels with large cross-sectional (*N*) and short and fixed time (*T*) dimensions, which hold in our case with $N = 58$ and $T = 30$. The green technology diffusion—the dependent variable in our study—is largely driven by general technology diffusion and



innovation. Therefore, some of the regressors, such as the per capita income, $CO_2$ emissions, environmental performance, technological achievement, and foreign direct investment, potentially depend on green technology diffusion. None of the other approaches based on the common correlated effects estimator of Pesaran [91] are appropriate in our case since they do not account for variable endogeneity. These alternative approaches work better for long $T$ and large $N$ and at best require weakly exogenous regressors (see e.g., Chudik and Pesaran [92] and Baltagi [90]). As Sarafidis [93] and Sarafidis and Wansbeek [94] point out, the GMM estimator can eliminate cross-sectional dependence and maintain its consistency. Moreover, a consistent GMM estimator is feasible using a subset of the instruments based on exogenous instruments (Sarafidis [93] and Sarafidis and Wansbeek [94]). Indeed, following Sarafidis et al. [95], we perform cross-sectional dependence tests on the residuals of the system GMM estimates (denoted SYR-CD), which are reported in Table 7. The SYR-CD test results show that the system GMM estimator accounts for all the cross-sectional dependence in the data, and our results do not suffer from cross-sectional dependence.

The estimates are obtained using a two-step system GMM since standard errors from a single-step system GMM are always asymptotically inefficient. In system GMM, the dependent variable's second to third lags and the first differences of all explanatory variables except lagged dependent variables are employed as instruments for the first differenced equation. The level equation is estimated using the first differences of endogenous variables as instruments. Due to the concerns of multicollinearity, autocorrelation, and endogeneity, we do not include all possible variables in each model; rather, we include a subset, as described in Table 1. The level and first-differenced models are assessed using the Lagrange multiplier Arellano-Bond autocorrelation tests for accurate dynamic specification and the Sargan-Hansen *J*-test for instrumental variable validity.

For the dynamic system GMM estimation with one lag, we should reject the first order serial correlation by the Lagrange multiplier Arellano-Bond test—LM-AR(1)—and not reject the second order serial correlation AR(2) using the LM-AR(2) test. The LM-AR(1) tests' results in Table 7 are all significant at the 1% significance, confirming the AR(1) specification. However, some of the LM-AR(2) tests are significant at the 5% level (Models 1–3, 6–8). For these cases, we estimate the models with two lags and find that both LM-AR(2) and LM-AR(3) tests are insignificant, invalidating the AR(2) specification. Therefore, all models are estimated with one lag of the dependent variable. Additionally, the Sargan-Hansen *J*-test results for the validity of over-identification constraints in Table 7 do not reject the null hypothesis of valid over-identification restrictions at all commonly used significance levels for all models that we estimate. As a result, we conclude that the specified instrumental variables are valid and that an AR(1) dynamic specification is sufficient to capture autocorrelation. The SYR-CD cross-sectional dependence test of Sarafidis et al. [95], reported in Table 7, which is valid for GMM estimates, indicates that the system GMM estimator adequately accounts for cross-sectional dependence in the data. As a result, our findings are not subject to cross-sectional dependence.

We estimate the eleven distinct models specified in Table 1 to evaluate the effects of the variables examined in this study as potential determinants of green technology diffusion. Our approach is to estimate regression models with key variables from Table 1, such as democratic accountability, foreign direct investment, socioeconomic conditions, and income distribution, while also developing instruments with the same subset. The estimates are summarized in Table 7. The empirical data indicate that the democratic accountability variable has a positive sign and is statistically significant across all specifications at the commonly used 10%, 5%, and 1% significance levels. Thus, more democratic conditions facilitate the adoption of green technologies, a finding that corroborates Zecca and Nicolli's [15] empirical evidence. Similarly, except for models 1 and 4, the effect of foreign direct investment is large with a positive sign in all other models. This empirical conclusion corroborates Halkos and Skouloudis's [38] finding that FDI has a beneficial influence on green technology development in a static panel data model. As a result, this

finding demonstrates the critical importance of foreign direct investment in a country's spread and development of green technologies.

The education index is closely linked with a number of other factors, including the TAI, per capita GDP, environmental performance index, and $CO_2$ emission. This is because education is one of the variables used to construct environmental performance and technological achievement indices. The association between education level and GDP is well established in the literature on economic growth (e.g., [96,97]). Additionally, a few studies establish an empirical link between education and $CO_2$ emission (e.g., [98–100]). Taking this into consideration, we include education in Models 1,3,5, and 7, but not in others, in order to avoid the identification problem that emerges when education is a sub-component of another variable, or multicollinearity. With the exception of Model 5, where education has a positive effect on green technology dissemination but is not statistically significant, the education index is statistically significant at all usual significance levels and has a positive sign in all models where education is included. Thus, it is recognized that a country's education level promotes the dissemination and development of green technology, most likely with a greater influence when combined with other encouraging elements. This finding is consistent with Bartel and Lichtenberg's [25] finding that highly educated individuals have a comparative advantage in adapting to and learning new technologies, which results in an increase in demand for new technologies.

According to Table 7, the socioeconomic conditions variable has positive coefficients and is statistically significant in all models except for Model 4, where the parameter estimate is statistically insignificant. As the empirical evidence indicates, socioeconomic conditions in countries have a positive effect on green technology diffusion. To gain a better understanding of the diffusion of green technology, we take another underlying variable into account: income inequality. In this study, inequality is quantified by the income share of the top 1% of highest-income households, denoted by the variable TOP1. In all models in which it is included, the inequality variable has a significant estimate at the 1% level and a negative sign, with the exception of Model 2, where the coefficient estimate is statistically insignificant. This finding is consistent with Vona and Patriarca's [13] empirical findings. Thus, the findings imply that a more equitable income distribution fosters green innovation in societies, while increasing economic inequality retards the diffusion and development of green technologies.

The $CO_2$ emission variable is included in models 2, 7, 9, and 10, but not in others due to its strong association with the education index, per capita GDP, and the TAI. $CO_2$ emission has a positive and statistically significant effect on the diffusion of green technologies in each specification at the 5% level of significance. As a result, in countries with larger $CO_2$ emissions, the diffusion of green technologies is enhanced. According to Alataş [101], environmental technologies have a statistically insignificant positive influence on $CO_2$ emission in the EU15's transport sector. In comparison to our findings, this is a reverse causality, albeit a statistically insignificant one. Due to the fact that we employ system GMM estimation, this reverse causality does not affect our estimates. However, Du et al. [14] find contradictory evidence. They demonstrate that whereas green technology advances have a negligible influence on $CO_2$ emissions in less developed economies, they have a considerable impact on $CO_2$ emissions in industrialized ones. However, this conclusion contradicts the findings of Naseem and Guang Ji [102], who find a statistically significant negative relationship between renewable energy usage and $CO_2$ emissions.

Due to the substantial association between education index and per capita GDP, the EPI is included in Models 3, 7–11, but not in others to avoid potential identification and multicollinearity. The results reveal that the EPI is statistically significant at the 1% level of significance in each model. Thus, our study establishes that in those countries where environmental performance is high the diffusion of green technologies is also stronger, indicating that prioritizing environmental issues causes faster adaptation of green technologies. GDP per capita is included in models 3, 9, and 11 because of its strong association with the education index, $CO_2$ emission, environmental performance index, and socioeconomic

conditions index. Except for Model 11, in which the effect of GDP is shown to be statistically insignificant, the GDP variable shows a positive sign and statistical significance at usual significance levels in relation to the diffusion of green technologies. It is inferred that countries with a higher per capita income will invest more in green technology and inventions, as well as file more patent applications. The conclusion is consistent with Vona and Patriarca [13] and Zecca and Nicolli [15], who demonstrate a positive and statistically significant association between GDP and environmental technology. Thus, adaptability and dissemination of green products and technologies are related to the per-capita income level and income distribution in all countries. Models 5, 10, and 11 include the general technology diffusion variable. The coefficient estimates for the general technology diffusion is positive and significant at the 1% level in all model specifications. Thus, the empirical evidence supports the notion that green technologies have a strong relationship and increase concurrently with the general diffusion of technology.

Lastly, the TAI, which is calculated by the authors, appears in Models 6 and 8, and is excluded from others due to its high correlation with the education, $CO_2$, GDP, and general technology diffusion variables. The estimates in Table 7 reveal that the technology achievement index has a positive effect on green technology diffusion, which is significant at the 1% level. Green technology adoption occurs more rapidly in countries with a high level of TAI because these countries have a track record of easily resolving environmental issues, minimizing environmental degradation, and enhancing environmental quality.

In summary, green technology diffusion is related to a society's per capita income, education level and investment profile, socioeconomic conditions and democratic accountability, $CO_2$ emissions, environmental performance, and general technology diffusion and innovation adaption. As a result, we use the system GMM technique to estimate parameters for various model specifications while taking into account the relationship between green technology diffusion and its determinants. As demonstrated in the table, adaption to green technology diffusion is positive and significant for all variables except inequality. The TOP1 variable's coefficient is statistically significant (except in model 2) and has a negative sign. This finding indicates that the diffusion of green technologies is strongly tied to the income distribution among countries. Additionally, among the 11 models, the most statistically significant specifications of model 3, model 6, and model 8 might be picked. All parameter coefficients are expected and statistically significant at the 1% level of significance in these models. The technical achievement score and the environmental performance index, in particular, are significant and favorably connected with the dissemination of green technologies.

In summary, green technology diffusion is related to the per capita income, education level, and investment profile of a country. Socio-economic conditions and democratic accountability, $CO_2$ emissions, environmental awareness, and adaptation of general innovations are also significant determinants of green technology diffusion. We estimate various model specifications using the system GMM method considering the interaction between green technology diffusion and its determinants, which may lead to endogeneity issues. Our estimates show that green technology diffusion relates positively and significant to all variables except for income inequality, for which the relationship is negative and significant. This conclusion demonstrates a substantial relationship between green technology dissemination and income distribution across countries, which is one of the study's novel results. Two further novel findings concern the technical achievement index and the environmental performance index, both of which are examined for the first time in our study. We demonstrate that these variables are major predictors of green technology diffusion. With rare exceptions, the majority of parameter estimates in the 11 models we estimate are statistically significant at traditional significance levels.

**Table 7.** The parameter estimates for various model specifications.

| Variable | Model 1 | Model 2 | Model 3 | Model 4 | Model 5 | Model 6 | Model 7 | Model 8 | Model 9 | Model 10 | Model 11 |
|---|---|---|---|---|---|---|---|---|---|---|---|
| L.LETD | 0.582 *** | 0.546 *** | 0.544 *** | 0.544 *** | 0.237 *** | 0.536 *** | 0.535 *** | 0.533 *** | 0.515 *** | 0.258 *** | 0.246 *** |
| | (0.030) | (0.021) | (0.011) | (0.024) | (0.015) | (0.012) | (0.014) | (0.017) | (0.027) | (0.018) | (0.019) |
| LDA | 0.075 *** | 0.140 *** | 0.068 ** | 0.093 *** | 0.038 * | 0.075 *** | 0.057 * | 0.068 ** | 0.081 *** | 0.057 *** | 0.034 * |
| | (0.022) | (0.024) | (0.023) | (0.012) | (0.015) | (0.021) | (0.024) | (0.024) | (0.012) | (0.011) | (0.015) |
| LEDU | 2.138 *** | | 1.852 *** | | 0.064 | | 1.753 *** | | | | |
| | (0.087) | | (0.097) | | (0.106) | | (0.106) | | | | |
| LFDIR | 0.006 | 0.014 *** | 0.011 ** | 0.004 | 0.005 * | 0.006 * | 0.011 ** | 0.014 *** | 0.007 * | 0.006 * | 0.007 ** |
| | (0.003) | (0.003) | (0.004) | (0.003) | (0.002) | (0.003) | (0.004) | (0.003) | (0.003) | (0.003) | (0.002) |
| LSOC | 0.134 *** | 0.242 *** | 0.145 *** | 0.020 | 0.044 *** | 0.197 *** | 0.160 *** | 0.195 *** | 0.046 ** | 0.037 ** | 0.035 ** |
| | (0.019) | (0.029) | (0.021) | (0.012) | (0.010) | (0.020) | (0.022) | (0.022) | (0.016) | (0.012) | (0.012) |
| LTOP1 | −0.249 *** | 0.002 | −0.257 *** | −0.270 *** | −0.218 *** | −0.230 *** | −0.245 *** | −0.223 *** | −0.289 *** | −0.244 *** | −0.211 *** |
| | (0.027) | (0.030) | (0.023) | (0.033) | (0.035) | (0.030) | (0.023) | (0.023) | (0.033) | (0.047) | (0.039) |
| LCO$_2$ | | 0.227 ** | | | | | 0.083 | | 0.015 | 0.001 | |
| | | (0.073) | | | | | (0.045) | | (0.078) | (0.033) | |
| LEPI | | | 0.335 *** | | | | 0.355 *** | 0.663 *** | 0.176 *** | 0.232 *** | 0.192 *** |
| | | | (0.032) | | | | (0.034) | (0.035) | (0.030) | (0.039) | (0.044) |
| LGDP | | | | 0.580 *** | | | | | 0.548 *** | | 0.033 |
| | | | | (0.018) | | | | | (0.034) | | (0.028) |
| LGTD | | | | | 0.566 *** | | | | | 0.564 *** | 0.550 *** |
| | | | | | (0.008) | | | | | (0.007) | (0.012) |
| LTAI | | | | | | 0.635 *** | | 0.458 *** | | | |
| | | | | | | (0.042) | | (0.037) | | | |
| Constant | 1.321 *** | 0.537 *** | 0.038 *** | −4.794 *** | −2.021 *** | 1.530 *** | −0.263 *** | −1.378 *** | −5.232 *** | −3.093 *** | −3.014 *** |
| | (0.051) | (0.097) | (0.190) | (0.179) | (0.096) | (0.099) | (0.212) | (0.153) | (0.327) | (0.214) | (0.309) |
| N | 1682 | 1682 | 1682 | 1682 | 1682 | 1682 | 1682 | 1682 | 1682 | 1682 | 1682 |
| $\hat{\sigma}^2$ | 0.279 | 0.271 | 0.268 | 0.268 | 0.166 | 0.269 | 0.266 | 0.268 | 0.260 | 0.170 | 0.167 |
| $\chi^2$ | 2239.752 *** | 2976.453 *** | 4325.612 *** | 5742.501 *** | 70672.990 *** | 20843.800 *** | 4251.478 *** | 18074.680 *** | 4778.272 *** | 23848.040 *** | 26126.06 ***0 |
| LM-AR(1) | −4.613 *** | −4.637 *** | −4.570 *** | −4.665 *** | −4.418 *** | −4.588 *** | −4.526 *** | −4.618 *** | −4.617 *** | −4.383 *** | −4.430 *** |
| LM-AR(2) | 2.458 ** | 2.387 ** | 2.378 ** | 2.408 * | 1.872 * | 2.389 ** | 2.350 ** | 2.387 ** | 2.363* | 1.944 * | 1.910 * |
| *J*-stat. | 56.914 | 56.885 | 57.431 | 55.380 | 50.096 | 56.516 | 56.501 | 55.739 | 56.406 | 47.691 | 49.086 |
| SYR-CD | 1.672 | 1.285 | 0.501 | 1.032 | 1.285 | 0.599 | 0.055 | 1.020 | 0.251 | 0.055 | 1.349 |

Note: The table reports system GMM estimates with Windmeijer-corrected standard errors in parentheses. *N* denotes number of observations, $\hat{\sigma}^2$ denotes residual variance, $\chi^2$ denotes Chi-square statistic for joint significance of all slope parameters, and LM-AR(1) and LM-AR(2) denote the Arellano-Bond test for first and second order serial correlation in the first-differenced residuals, respectively. Sargan *J* stat is the Sargan test of the overidentifying restrictions. SYR-CD is the cross-sectional dependence test of Sarafidis et al. [95]. *, **, and *** denote significance at 10%, 5%, and 1% levels, respectively.

*4.2. Discussion*

The determinants of green technology diffusion were examined empirically in the context of the general technology diffusion trends, environmental performance, democratic accountability, income distribution, income level, socioeconomic conditions, and technological achievement of nations using a large panel of 58 nations. Our empirical results indicate that the relationships between green technology diffusion and income level, education level, investment profile, socioeconomic conditions, democratic accountability, $CO_2$ emission, environmental performance, brow technology diffusion, and technological achievements are statistically significant and positive in selected countries. Thus, improvements in these factors help increase the adaptation and spread of green technologies. Firstly, the income level of countries is a predominant factor for enhancing green products and sustainable development. Fatima et al. [103] find that the ratio of consumption of renewable energy to $CO_2$ emission descends when income rising. At the same time, some studies [15,16,104] conclude that higher GDP increases carbon emissions. Secondly, the investment profiles of countries play an important role in the adaption of green innovations. Khan et al. [105] show significant causal relationships between policies of exports and imports, income level, and green innovation that have resulted in changes to consumption-based $CO_2$ emission levels in G7 countries. Likewise, Shahzad et al. [106] find that export diversification greatly decreases $CO_2$ emissions in selected developed and developing countries. These outcomes are comparable to the conclusion of Andersson [107] on Chinese exports. In this respect, our results are complimentary to the empirical evidence presented in these studies, but in a more extensive coverage of countries, which includes both developing and developed countries.

Thirdly, economic development, socio-economic conditions, democratic accountability, technological innovation, and environmental performance are particularly prominent factors shaping green technologies. Wang et al. [108] indicate that political freedom and institutional quality help to reduce $CO_2$ emission levels when GDP growth and financial development are enhancing environmental degradation. The result is in line with the findings of [109,110]. Zuhair and Kurian [111] recognize some socio-economic barriers which affect the diffusion of green products, such as the gender gap, lack of human and financial capacity, loss of community spirit, and lack of environmental awareness. Moreover, Chaudhry et al. [112] find that technological innovations have a significantly positive relationship with environmental indicators in higher-income East-Asian and Pacific countries.

Energy-efficient and environmentally friendly technologies, as well as patents, promote environmental technology development. Paramati et al. [113] show that green technology helps to decrease energy consumption and improve energy efficiency. These results are similar to the outcomes of [114,115]. Another study by Arbolino et al. [116] concludes that economic variables have an important role in the diffusion of green technology policies by using the EPI.

Overall, our empirical analysis demonstrates how the determinants of green technology diffusion make a positive contribution to attaining a sustainable environment as described in the relevant literature. However, our results go beyond the previous studies by including a larger set of factors that affect environmental technology diffusion. Our study obtains complimentary evidence to previous literature from a broader range of time periods and a larger number of countries that includes both developing and developed economies. More importantly, our study is the first to obtain evidence that green technology does not independently develop and diffuse from the general or brown technology trends and technological achievement of countries. The general technology diffusion and achievement trends in a country are significant drivers of the green technology trends.

## 5. Concluding Remarks and Policy Implications

The main objective of this paper is to identify the major economic, social, political, social, and environmental determinants of green technology diffusion. We consider per capita income, income inequality, education level, foreign direct investment, socio-economic conditions, democratic accountability, $CO_2$ emissions, environmental performance, general technology diffusion trends, and technology achievement index of nations as potential factors affecting green technology diffusion. We use the two-step system GMM estimation method proposed by Arellano and Bover [63] and Blundell and Bond [64] to estimate various dynamic panel data models. An annual frequency panel dataset for the period of 1990–2019 is used to examine the links between green technology diffusion and its explanatory factors. Our empirical findings show that inequality has a statistically significant and negative relationship with green technology diffusion. However, other independent and instrumental variables have a significant and positive correlation with the adaptation process to green technologies across countries over the period from 1990 to 2019.

In the light of our findings, it appears that macro conditions at country levels may impede the promotion of green technology development and diffusion. Several of our findings require careful consideration. Our findings suggest that the Popper hypothesis in favor of democracy's favorable impact in environmental technology innovation also applies to environmental technology diffusion. All specifications covered in this paper exemplify democracy's proactive role. The findings of country-level empirical research conducted using dynamic GMM estimation demonstrate that increased educational attainment has an independent, positive effect on the diffusion of green technologies. This is true after controlling for a variety of other variables, such as income, socioeconomic conditions, and so on. These findings demonstrate that improving environmental conditions amplify the effect of increasing educational attainment on labor productivity and economic well-being. This approach could be used in conjunction with or in substitution of already available methods of sustainable development. Additionally, our findings indicate that there are

increased benefits associated with more equitable income distribution and environmental performance enhancements, as they amplify the adoption of environmentally friendly technologies that promote sustainable development and aid in mitigating the effects of climate change. More importantly, our empirical results have some more important implications.

Our findings have implications for policy makers. We find that technological achievement and trends in the diffusion of brown technologies are important drivers of green technology diffusion. If the green technologies are not adopted through widespread diffusion, the environmental benefits of environmental technology innovations do not realize. Therefore, policies such as public R&D activity spending, environmental tax implementations, preferential tariffs, investment incentives, voluntary programs, and environment certificates that aim to help companies adopt green technologies and produce in an environmentally friendly manner may not be successful, because national-level factors such as technological achievement, human capital, and general technology diffusion are the main drivers of green technology diffusion.

Our findings suggest that national policies promoting brown technology innovation and diffusion benefit not only brown technology, but also contribute to climate change mitigation through green technology diffusion. Likewise, this holds true for a nation's technological achievement. Thus, investing in technology may be a less distorting option than enacting climate-related regulations and rules. In this sense, our findings imply that environmental conservation is a significant additional social externality associated with investments in both green and brown technologies. Global and national policymakers must gain a better understanding of the critical determinants defining national terrains that affect the effective development and diffusion of environmental technologies. This is especially critical for less developed countries, as assessments can help to inform the portfolio of structural adjustment policies. Policymakers should consider carefully whether macroeconomic factors could result in a rebound effect or a lock-in mechanism. As a result, they should prioritize non-price policies over carbon taxation. Increased incentives and subsidies should be used to accelerate the adoption of green technologies.

Finally, it may be worthwhile to investigate the phenomenon from a variety of angles, employing appropriate variables at various levels of analysis and examining their interactions. Clearly, dynamic specifications applied to longer time series can assist in this endeavor by examining the rate of diffusion and the effect of time on the development and dissemination of environmental technology. Our findings, we believe, establish a foundation for a more nuanced understanding of the factors influencing country-level patterns in a variety of technology domains.

**Author Contributions:** Conceptualization, M.B.; methodology, M.B.; software, B.A. and M.B.; validation, M.B.; formal analysis, B.A. and M.B.; investigation, B.A.; resources, B.A. and M.B.; data curation, B.A.; writing—original draft preparation, B.A. and M.B.; writing—review and editing, B.A. and M.B.; visualization, B.A.; supervision, M.B.; project administration, M.B. All authors have read and agreed to the published version of the manuscript.

**Funding:** This research received no external funding.

**Institutional Review Board Statement:** Not applicable.

**Informed Consent Statement:** Not applicable.

**Data Availability Statement:** Data and material are available from the authors upon request.

**Conflicts of Interest:** The authors declare no conflict of interest.

## Appendix A

List of countries included in the data set.

| | | |
|---|---|---|
| Argentina | Hungary | Portugal |
| Australia | Iceland | Romania |
| Austria | India | Russia |
| Belarus | Ireland | Saudi Arabia |
| Belgium | Israel | Serbia |
| Brazil | Italy | Singapore |
| Bulgaria | Japan | Slovakia |
| Canada | Kazakhstan | Slovenia |
| China | Latvia | South Africa |
| Colombia | Lithuania | Spain |
| Croatia | Luxembourg | Sweden |
| Cyprus | Mexico | Switzerland |
| Czechia | Moldova | Thailand |
| Denmark | Morocco | Turkey |
| Egypt | Netherlands | United Kingdom |
| Estonia | New Zealand | United States |
| Finland | Norway | Uruguay |
| France | Peru | Venezuela |
| Germany | Philippines | |
| Greece | Poland | |

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
