# Peer review of "On the Determinants of Green Technology Diffusion: An Empirical Analysis of Economic, Social, Political, and Environmental Factors"

_sustainability, doi:10.3390/su14042008_

Round 1

Reviewer 1 Report

GREEN TECHNOLOGY DIFFUSION FOR SUSTAINABLE DEVELOPMENT: AN EMPIRICAL ANALYSIS CAUSAL FACTORS

In this manuscript, the authors addressed an interesting issue. I think the scientific depth and clarity of this manuscript can be significantly enhanced, if the authors make the following changes,
 1- Please revise the title

2-The abstract doesn’t express the emphasis. It needs to be modified.

3-The authors did not mention clearly their objective in the introduction and add to the contribution of the study.

4- The discussion of outcomes can further be improved by comparing the latest studies.
5- More policy implications are needed in the conclusion section, by including the limitation of the study.

6- CO2 abrebivations should be fixed in the whole text. It must be subscript 

7- The manuscript didnot used the referecing style of Sustaianbility journal 

8- Why the authors used the  GMM appraoch rather than CS-ARDL, AMG, CCE or DYNAMIC CCE approaches? 

Author Response

Responses are given in the attached PDF file.

Reviewer 2 Report

Review Report on Research Article tittle “ Green Technology Diffusion for Sustainable Development: An Empirical Analysis Causal Factor

Manuscript ID: 1558005

I have looked at Research Article for evaluation. Some suggested changes:

  1. Abstract: Abstract is not giving clear picture of study objective -- please make it clear and systematic with objective and results of the study
  2. Introduction: Knowledge gap is missing-please add at least one knowledge gap for topic and objective of the study are very general and too long statements,-convert these objectives into small, simple and clear statements

Please update the literature and relate it with recent studies; such as (https://doi.org/10.1016/j.rser.2021.111735 and https://doi.org/10.1016/j.renene.2021.10.084)

  1. Methodology sections is well written but only need to mention the selection of n=58 countries, on what basis you have selected these countries, briefly explain
  2. Results: there are several abbreviations without full explanations (such as OECD)-recheck all and avoid small errors
  3. Please add discussion section and compare your results with some of these below studies;
  • Export product diversification and CO2 emissions: Contextual evidences from developing and developed economies
  • Renewable and nonrenewable energy consumption, trade and CO2 emissions in high emitter countries: does the income level matter?
  1. Conclusion: starting line of conclusion section is “the paper investigate the impact of green technology on economics, social factors” while in introduction/abstract it seems that this study is exploring determining factors of green technology diffusion- cross check this, make clear statement and be consistent throughout the article writing
  2. References: Check the reference cited section and develop it in accordance with the journal requirements and format (of sustainability).Further, there is a lot of repetition for same reference even within a same paragraph (such as Allan, et al., 2014; Waisman et al., 2019), try to avoid this nature of writing and add new reference

Author Response

(The authors gave the same response as above.)

Round 2

Reviewer 1 Report

the manuscript can be accepted since the all suggestions are done